# Novel Neuroprotective Agents to Treat Neonatal Hypoxic-Ischemic Encephalopathy: Inter-Alpha Inhibitor Proteins

**DOI:** 10.3390/ijms21239193

**Published:** 2020-12-02

**Authors:** Liam M. Koehn, Xiaodi Chen, Aric F. Logsdon, Yow-Pin Lim, Barbara S. Stonestreet

**Affiliations:** 1Department of Pediatrics, The Alpert Medical School of Brown University, Women & Infants Hospital of Rhode Island, Providence, RI 02905, USA; lkoehn@wihri.org (L.M.K.); xchen@wihri.org (X.C.); 2Geriatrics Research Education and Clinical Center, Veterans Affairs Puget Sound Health Care System, Seattle, WA 98108, USA; alogsdon@uw.edu; 3Department of Medicine, Division of Gerontology and Geriatric Medicine, University of Washington School of Medicine, Seattle, WA 98195, USA; 4ProThera Biologics, Inc., Providence, RI 02903, USA; yplim@protherabiologics.com; 5Department of Pathology and Laboratory Medicine, The Alpert Medical School of Brown University, Providence, RI 02903, USA

**Keywords:** asphyxia, blood-brain barrier, brain, development, encephalopathy, hypoxia-ischemia, inflammation, inter-alpha inhibitor proteins, neonatal, neuroprotection

## Abstract

Perinatal hypoxia-ischemia (HI) is a major cause of brain injury and mortality in neonates. Hypoxic-ischemic encephalopathy (HIE) predisposes infants to long-term cognitive deficits that influence their quality of life and place a large burden on society. The only approved treatment to protect the brain after HI is therapeutic hypothermia, which has limited effectiveness, a narrow therapeutic time window, and is not considered safe for treatment of premature infants. Alternative or adjunctive therapies are needed to improve outcomes of full-term and premature infants after exposure to HI. Inter-alpha inhibitor proteins (IAIPs) are immunomodulatory molecules that are proposed to limit the progression of neonatal inflammatory conditions, such as sepsis. Inflammation exacerbates neonatal HIE and suggests that IAIPs could attenuate HI-related brain injury and improve cognitive outcomes associated with HIE. Recent studies have shown that intraperitoneal treatment with IAIPs can decrease neuronal and non-neuronal cell death, attenuate glial responses and leukocyte invasion, and provide long-term behavioral benefits in neonatal rat models of HI-related brain injury. The present review summarizes these findings and outlines the remaining experimental analyses necessary to determine the clinical applicability of this promising neuroprotective treatment for neonatal HI-related brain injury.

## 1. Introduction

Reductions in oxygenated blood flow to the fetal or neonatal brain can result in mortality or severe long-term cognitive impairment. Hypoxia-ischemia (HI) limits nutrient and energy supply to the brain and stimulates brain parenchymal and systemic inflammatory responses [1,2,3,4]. These abnormalities predispose patients to neurological cell death, cellular dysfunction, and neurodevelopmental deficits, which comprise hypoxic-ischemic encephalopathy (HIE). HIE can develop because of complications before or during birth, including maternal vascular disorders, uterine rupture, umbilical cord entrapment, placental insufficiency, and pre-eclampsia. Neonatal cardiopulmonary abnormalities after birth, including hemorrhage, pulmonary disorders, or apnea, can also predispose infants to hypoxic- or HI-related brain injury. Although these events can occur in any pregnancy, very premature infants are at particularly high risk for disorders that predispose them to neurodevelopmental abnormalities [5,6]. Cognitive disabilities not only impact an individual’s quality of life but also come with large economic burdens for families and the health care and educational systems. Necessary care costs are estimated to exceed $1 million per patient over the course of a lifetime [7].

Currently, therapeutic hypothermia is the only clinically approved neuroprotective treatment strategy for newborns exposed to HI [8]. However, this therapy is not approved to treat premature infants, the population with the highest incidence of HI-related brain injury and abnormal neurodevelopmental outcomes. Therapeutic hypothermia is only partially protective in full-term patients [9,10,11,12] and has a narrow therapeutic window because it is most efficacious when administered within 6 h of birth [13,14,15]. Therefore, there is a need for novel alternative or adjunctive therapeutic options that can provide the best possible preservation of neurological tissue for neonatal patients exposed to HI.

## 2. Inter-Alpha Inhibitor Proteins

### 2.1. Structure and Function

Inter-alpha inhibitor proteins (IAIPs) are a group of structurally related, naturally occurring molecules that are present in high concentrations in human plasma of premature and full-term infants, and in adults [16,17,18,19,20,21,22,23,24]. They are also present in many cell types throughout the body, including neurons and glia in the brain [25,26]. The major forms of IAIPs in plasma are inter-alpha inhibitor (*IaI*), which has two heavy chains (HC1 and HC2) and one light chain (LC), and pre-alpha inhibitor (*PaI*), with one heavy (HC3) and one light chain (LC); see Figure 1 [16]. Chondroitin sulfate connects the IAIP light and heavy chains in the form of a unique protein-glycosaminoglycan-protein crosslink [27,28,29]. The LC, also known as bikunin, ulinastatin, or urinary trypsin inhibitor, contains a serine protease inhibitor core (kunitz-type domain) [30], whereas the heavy chains mainly function to stabilize the extracellular matrix [21,31]. IAIPs, in the collective sense, are immunomodulatory molecules that limit protease induced damage, inhibit pro-inflammatory cytokines, augment anti-inflammatory cytokines, alter immune-cell morphology/response, reduce complement activation, and bind to histones during systemic inflammatory conditions [16,32,33,34,35,36,37]. These functions make IAIPs promising drug targets for the treatment of conditions, in which excessive inflammatory responses may contribute to disease progression.

### 2.2. Treatment of Inflammatory Conditions

Systemic IAIP levels are reduced in neonatal sepsis [24,32] and necrotizing enterocolitis (NEC) [38], suggesting that these proteins may be decreased during systemic inflammation in premature infants. The liver also reduces the production of LC and H2, which limits the replacement of circulating IAIPs during the acute phase of inflammation [39]. Replenishing IAIP levels with exogenous drug treatment could provide the body with the reserves required to effectively attenuate excessive inflammation. Highly efficient extraction and purification processes from human plasma have been successfully developed because both major forms of IAIPs (*IaI* and *PaI*) are found in relatively high concentrations in human blood. Both the 250 kDa *IaI* and 125 kDa *PaI* have been co-purified and isolated together during chromatographic separation on anion-exchange and proprietary affinity synthetic mimetic columns [40,41,42]. The highly purified IAIPs are biologically active as assessed by quantitatively measuring the ability of the proteins to inhibit the hydrolysis of a chromogenic trypsin substrate and monitoring the rate of decrease on a spectrometer [43].

The therapeutic potential of treatment with the human plasma derived IAIPs to limit inflammatory damage has been demonstrated in murine models of anthrax exposure [44,45] and in both adult [46,47] and neonatal [35] rodents that have been exposed to sepsis and inflammatory stimuli, such as lipopolysaccharide (LPS). In contrast to the complete complex molecules of IAIPs that are extracted from human plasma, the light chain subunit of IAIPs, bikunin, that is excreted mainly by the kidneys, has been isolated from human urine and shown also to be protective in models of HI and inflammation related disorders [48,49,50,51]. The therapeutic potential of bikunin to attenuate key stages in the progression of cerebral HI-induced injury in adult subjects has recently been reviewed [52]. Bikunin therapy has been approved for clinical use in Japan and China to treat inflammatory conditions, such as pancreatitis [53]. However, due to its short plasma half-life (<10 min), continuous intravenous administration would be required for extended treatments [54]. Pharmacokinetic analysis of IAIPs in neonatal HI models has shown a much longer half-life of 16–23 h [41]. The complete molecular form of IAIPs may be more appropriate agents to treat neonatal subjects exposed to HI, in which brain injury evolves over a period of days to weeks [4,55]. Furthermore, IAIP modulation of certain immune cells differs considerably from that of bikunin, which may provide additional therapeutic benefits [36].

### 2.3. Applicability of IAIPs to Hypoxic-Ischemic Encephalopathy

Systemic IAIP levels are reduced after neonatal HI, providing similarities to the IAIP plasma profiles observed in neonatal inflammatory conditions, including sepsis and NEC [32,38]. Sepsis and NEC predispose premature infants to brain injury [56,57], raising the interesting possibility that reductions in IAIPs may be a contributing factor to brain injury in preterm infants [56,57]. Neuroinflammation is a major contributor to the evolution of HI-related brain injury during development [4,40,58]. Pro-inflammatory cytokines and cytotoxic factors released by glial and immune cells contribute to HI-related neuroinflammatory responses that can result in neuronal loss and cognitive impairment [59,60]. Systemic cytokine levels have even been correlated with the severity of brain injury in infants exposed to HI [61]. The therapeutic potential of IAIPs in models of neonatal inflammation coupled with the role of inflammation in HIE-related brain injury suggests that IAIPs could be very promising therapeutic drug candidates for this disorder [35,60,62]. They may provide alternative or adjunctive therapeutic options to attenuate the severity of HIE in full-term infants or as potential neuroprotective agents for premature infants at high-risk for neurodevelopmental disabilities.

## 3. Neuroprotective Properties of IAIPs

### 3.1. Preservation of Brain Tissue

Recent studies have shown that treatment with human blood derived IAIPs can provide considerable neuroprotection in neonatal rat models of HI-related brain injury. Intraperitoneal IAIP injections (30–60 mg/kg) reduced the volume of infarcts in the neonatal brain by 35–50% [40,63]. In addition, treatment has been shown to preserve overall brain weight and attenuate histopathological injury within the cerebral cortex, hippocampus, caudate/putamen, cingulate, and thalamus [40,64]. These results suggest that systemic treatment with IAIPs can attenuate brain injury after exposure to HI in neonatal subjects.

### 3.2. Cellular Response

Treatment with IAIPs can limit HI-induced changes in the cellular composition of the brain, in addition to the gross morphological preservation. HIE is characterized by reductions in neurons and oligodendrocytes. Fluoro-Jade B staining [64] and ApopTag/neuronal nuclei (NeuN) co-localization [40] have shown that IAIPs can reduce neuronal cell death in the cerebral cortex of neonatal rats after exposure to HI. Treatment was also shown to preserve the area of oligodendrocyte staining (CNPase; 2′,3′-cyclic nucleotide 3′-phosphodiesterase) [40]. In contrast to neuronal and oligodendrocyte losses, HI results in an increase in microglia, astrocytes, and leukocytes in the brain. Treatment with IAIPs limits the increases in cerebral cortical astrocytes, hippocampal microglia, and corpus callosum infiltrating leukocytes in neonatal rats after exposure to IAIPs [65]. IAIPs can also play an important role in preventing damage to the pulmonary vascular endothelium in mice [66]. Consistent with these findings in the pulmonary vascular endothelium, treatment with IAIPs has recently been shown to protect against LPS-induced blood-brain barrier (BBB) disruption in adult mice [67]. However, the impact of exogenous IAIP treatment on the components of the neurovascular unit, including endothelial cells, astrocytes and pericytes, remains to be determined.

### 3.3. Neurobehavioral Effects

The preservation of neurological tissue and attenuation of cellular death in response to IAIP treatment is associated with long-term neurobehavioral benefits. Neonatal rats exposed to HI (8% O_2_, 2 h) that were treated with IAIPs (30 mg/kg/day, 2 doses) performed better in water maze tasks for non-spatial and spatial learning 5–11 weeks after exposure to HI [64]. Combining IAIP treatment with experience based learning in juvenile rats has also been shown to result in long-term (13–16 weeks post-injury) improvements in the efficiency of working memory [68]. Neonatal brain injury is known to result in auditory deficits that contribute to language comprehension and learning difficulties [69,70]. Rats treated with IAIPs after neonatal HI, when tested as adults (23 weeks after injury), exhibited significantly better auditory discrimination capacity compared with untreated counterparts after injury [71]. These results demonstrate that short-term treatment with exogenous IAIPs after HI in neonatal subjects has the potential to provide cognitive improvements that persist into later life.

## 4. Mechanism of Action

Despite studies suggesting promising neuroprotective effects of treatment with IAIPs, the specific mechanism(s) underlying these changes have not been completely elucidated. Figure 2 summarizes some potential mechanisms that are currently under consideration.

### 4.1. IAIP Transfer Into the Brain

IAIPs are relatively large, hydrophobic molecules suggesting that their permeability from the systemic circulation across the BBBs is most likely limited [72]. HI decreases tight junctions at the BBB and increases barrier permeability to hydrophobic compounds in the fetal and neonatal brain [73,74,75,76]. Nonetheless, the level of exogenously administered IAIPs that could cross the BBB under conditions of HI could remain low relative to the quantity of IAIPs endogenously present within the brain parenchyma [77,78]. The presence of high levels of endogenous IAIPs within the brain parenchyma raises the question of whether influx transport systems exist for either IAIPs or certain IAIP subunits [39,79]. It is possible that not all of the protein chains of IAIPs can be synthesized within the brain [39,79]. Determination of IAIP transfer across the intact BBB and at several time points after HI could clarify whether systemically administered IAIPs could be able to penetrate the BBB in appreciable quantities. If IAIPs were able to readily cross the BBB, they could potentially reduce the progression of neurological injury through direct cellular and/or molecular interactions within the brain parenchyma (Figure 2A). Conclusions regarding the transport of IAIPs across the BBB await further investigation using methods similar to those previously reported for cytokines [80,81].

### 4.2. Attenuating Immune Cell Infiltration

Recent work has suggested that IAIPs could act systemically to reduce the transfer of detrimental stimuli into the brain, thereby reducing brain damage. Treatment with IAIPs reduced myeloperoxidase (MPO) staining and co-localization of MPO and matrix metalloproteinase 9 (MMP9) in the brain of neonatal rats exposed to HI-related brain injury [65]. MPO is expressed mainly by neutrophils and to a lesser extent by other immune cells, including monocytes and macrophages [9,82,83]. MMP9 is an inflammatory factor released by neutrophils that is known to damage the BBB and increase neuronal cell death [84,85,86]. MMP9 serum levels are known to correlate with neurological abnormalities in infants that have been exposed to perinatal asphyxia [87]. Studies have shown that IAIPs can alter neutrophil morphology, decrease their adhesion to endothelial cells, and limit the release of damaging factors, such as reactive oxygen species [36]. In addition to acting on immune cells directly, studies in the pulmonary vasculature have shown that IAIPs can attenuate inflammation-induced leukocyte adhesion molecule expression [66]. This process may reduce the ability of immune cells to infiltrate tissues, including the brain. Therefore, the potential exists that systemic administration of IAIPs could decrease the quantity of immune cell activation and, thereby, reduce their transfer into the brain, where they could contribute to neurological damage (Figure 2B).

### 4.3. Reducing Damaging Mediators in the Circulation

Treatment with IAIPs could reduce the level of cytokines and other potentially harmful molecules within the systemic circulation. Cytokines are known to penetrate the BBB [80,81], and exposure to HI-related brain injury significantly increases cytokine transport into multiple brain regions [88]. Exogenous IAIP treatment may act systemically to reduce the level of circulating cytokines, thereby limiting the quantity that could enter the brain to cause injury (Figure 2C). Systemic treatment with IAIPs have been shown to reduce LPS-induced cytokines in the serum [35,67]. However, significant reductions in specific cytokines simultaneously in serum and brain have not yet been identified [67]. The potential exists that factors detrimental to the brain that have not been examined, including MMP9 or vascular endothelial growth factor, could have lower profiles in the systemic circulation and brain after systemic treatment with IAIPs. The release of reactive oxygen species by immune cells in the circulation decreases after treatment with IAIPs, which could also reduce the damage to the brain by limiting exposure to systemically produced molecules [36]. The potential also exists that exogenous treatment with IAIPs could attenuate signal transduction pathways at the vascular endothelial interface that could modulate the cells of the neurovascular unit to attenuate the relay of various inflammatory signals into the brain [89,90,91]. Experimentation analyzing inflammatory mediator responses in the serum and brain after exposure to neonatal HI is currently underway.

### 4.4. Preserving Blood-Brain Barrier Integrity

HI-related brain injury in the fetus and neonate is associated with increased BBB permeability [73,74,76]. Treatment with IAIPs could decrease the level of molecules in the systemic circulation known to disrupt BBB integrity (Figure 2D) [90]. LPS exposure increases serum levels of interleukin-6 (IL-6) which is known to disrupt normal BBB function [67,92,93]. Treatment with IAIPs attenuates LPS-induced BBB disruption and also reduces IL-6 in the systemic circulation of adult male mice [67]. Therefore, IAIPs may preserve BBB function through mechanisms associated with systemic cytokine suppression during inflammation-related conditions. Preservation of BBB integrity may reduce the transfer of inflammatory cytokines, endogenous plasma proteins, and other systemic molecules into the brain where they could potentially initiate injury. The effects of systemic treatment with IAIPs on the BBB are important because increased BBB permeability has been shown to impair cognition and accentuate the development of neurological disorders [94,95,96].

### 4.5. Molecular Interactions

IAIPs are known to participate in molecular interactions that transfer, displace, or replace heavy chains. Interactions with tumor necrosis factor stimulated gene 6 (TSG-6) have been associated with anti-inflammatory effects, including the inhibition of plasmin [97,98]. Complexes of hyaluronan and IAIP heavy chains have been linked to immune responses [31,99] and immune cell trafficking [31]. IAIPs have also been shown to co-localize with hyaluronan on the luminal surface of blood vessels in lung endothelium [66]. Therefore, the potential exists for fragments of IAIPs or molecular complexes of IAIP components to perform functions or initiate downstream cascades independently of the intact IAIP molecule (Figure 2E).

## 5. Future Directions

### 5.1. Sex-Specific Efficacy

Future studies are required to determine whether the efficacy of treatment with IAIPs is sex-specific. Initial studies identifying neuroprotection by IAIPs in neonatal rodents exposed to HI investigated mainly male subjects [64,68,71]. More recent studies investigating both male and female subjects suggest that, although brain infarct volumes were reduced in both sexes after treatment with IAIPs given soon after HI [40,63], when treatment was delayed, males exhibited greater reductions in infarct volumes than female neonatal rodents [63]. In addition, studies have shown that male rats exposed to HI and treated with IAIPs exhibit significant differences in pathological scoring, neuronal and non-neuronal cell death, and inflammatory responses compared with untreated male rats, whereas no significant differences were observed in female cohorts [40,65]. Treatment with IAIPs after exposure to LPS results in changes in serum cytokine levels and attenuation of increased BBB permeability preferentially in adult male mice [67]. The effects of IAIPs on male subjects are of clinical interest because human male neonates are known to have a higher incidence of brain injury and worse long-term cognitive outcomes compared with females [62,100,101]. An important area of future research would be to investigate modifications to the treatment regimen of IAIPs in order to improve efficacy in female subjects.

### 5.2. Optimal Treatment Regimens

The optimal treatment regimen and doses of IAIPs have not been established as yet. Thus far, intraperitoneal doses 30 and 60 mg/kg/day of IAIPs have been investigated in neonatal rodents exposed to HI and adult rodents exposed to LPS [40,63,64,65,66,67,68,71]. Recent pharmacokinetic analysis has shown that the plasma half-life of IAIPs is 9–12 h in control neonatal rats and 16–23 h in neonatal rats exposed to HI [41]. These results provide some evidence to support separating doses by 24 h in neonates exposed to HI-related insults. However, analyses of the direct neuroprotective effects of different doses and treatment regimen with IAIPs have not as yet been examined. Therefore, studies are required to examine different doses of IAIPs in animals exposed to the same severity of HI to determine the optimal doses of IAIPs. It also remains possible that doses lower than 30 mg/kg may also be efficacious or that a higher dose may be more effective at protecting the brain from HI-related insults. Safety analysis has suggested that 60 mg/kg of IAIPs do not produce adverse effects on body weight or bleeding times in neonatal rats exposed to HI, suggesting that high doses could be considered to improve efficacy [41]. Thirty or 45 mg/kg doses provided similar efficacy in neonatal rodents exposed to sepsis resulting in markedly improved survival, although doses of 15 mg/kg were not as effective [35]. Direct dose-response comparisons for the neuroprotective efficacy of IAIPs after exposure to neonatal HI remain to be performed. However, the 60 mg/kg IAIP treatment has been shown to reduce infarct volume by 50% even after exposure to 2 h of hypoxia [63], whereas 30 mg/kg treatment reduced infarct volume by 35% after 90 min of hypoxia [40]. It is also not known how long treatment may need to be continued in order to maximize the neuroprotective efficacy of IAIPs. Long-term behavioral analysis has identified that just two IAIP doses (zero h and 24 h after HI) were sufficient to produce long-term behavioral benefits [64,68,71]. It remains to be determined whether extending treatment for much longer than a two-day interval could be required to achieve maximum benefits because neurological cell death can continue to occur for days to weeks after neonatal brain injury [4,102]. The efficacy of IAIP treatment when delayed for one and six hours after neonatal HI has been investigated to study the therapeutic window of administration after injury [40,63]. IAIP treatment immediately after HI (zero h) decreased infarct volumes in the brain by approximately 50%, but, when treatment was delayed for one hour after injury, infarct was only reduced in males (24%) and not significantly reduced in females [63]. Delaying treatment 6 h after HI did not improve neuropathological scoring of the brain compared with placebo treatment but did preserve overall brain weight in males [40]. Therefore, IAIPs may have the capacity to protect the neonatal brain in male rats when administered up to six hours after injury. However, under current regimens, the greatest efficacy for both sexes is achieved when IAIP administration occurs within one hour of injury. Future studies are required to identify the optimal dose, frequency, and duration of IAIP administration after neonatal HI to maximize neuroprotection in both males and females and to advance this treatment into clinical application.

### 5.3. Comparison to Hypothermia

A comparative analysis between the novel IAIP drug treatment and the current clinically available treatment, therapeutic hypothermia, is yet to be available. In order for a new therapeutic treatment to make a successful transition to the clinical setting, the therapy must provide as much or more neuroprotection than the currently approved therapy. Such analysis should include morphological assessment of brain tissue preservation, analysis of cellular responses, and behavioral testing. Due to differences in animal models, ages, surgical procedures, and experimental measurements, it is currently not feasible to directly compare IAIPs and therapeutic hypothermia from separate studies available in the literature. However, there is evidence to suggest that the morphological and cellular neuroprotection provided by IAIPs [40,65] may not always be observed in hypothermic models [103,104,105]. These differences in treatment responses may suggest direct advantages of IAIPs over hypothermia or highlight how concurrent treatment of both therapies may result in maximum neuroprotection to protect the brain by means of different mechanisms.

## 6. Conclusions and Perspectives

Systemic administration of exogenous IAIPs is a promising neuroprotective strategy in neonates after exposure to HI-related brain injury. Not only does the treatment reduce the infarct volume in the brain and improve brain weight, but studies have also shown that IAIP treatment can attenuate specific cellular responses within the brain. Preservation of neurons and oligodendrocytes, along with reductions in microglia, astrocytes, and leukocytes, provides evidence that treatment is limiting the progression of neurological damage. Rodent behavioral analysis has shown that the neuroprotection provided by IAIPs translates to long-term improvements in cognitive abilities, such as auditory processing, working memory, and both spatial and non-spatial learning. Further research is still required to determine the optimal treatment regimen and neuroprotective mechanisms of IAIPs. Future analysis must consider the sex-specific differences in efficacy and determine how to ensure that both male and female patients can receive the maximum benefit of this novel therapy.

## Figures and Tables

**Figure 1 ijms-21-09193-f001:**
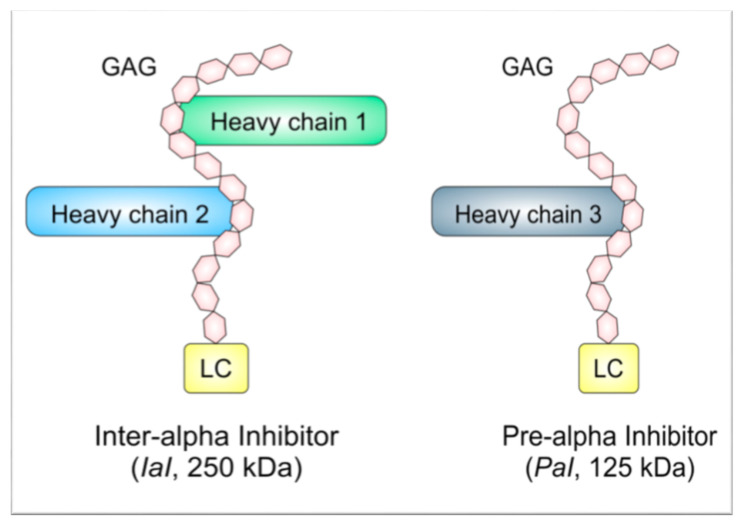
The structures of two major inter-alpha inhibitor proteins (IAIPs) endogenously present in serum, inter-alpha inhibitor (*IaI*; 250 kDa) and pre-alpha inhibitor (*PaI*; 125 kDa). IAIPs contain a glycosaminoglycan (GAG) backbone of chondroitin sulfate disaccharide repeats. Protein components, termed heavy chains, are connected to the GAG backbone. The light chain (LC; ~25 kDa) is present on both *IaI* and *PaI*. Heavy chains (HC; ~100 kDa) differ between IAIPs, with heavy chain 1 (HC1) and heavy chain 2 (HC2) on *IaI* and heavy chain 3 (HC3) on *PaI*.

**Figure 2 ijms-21-09193-f002:**
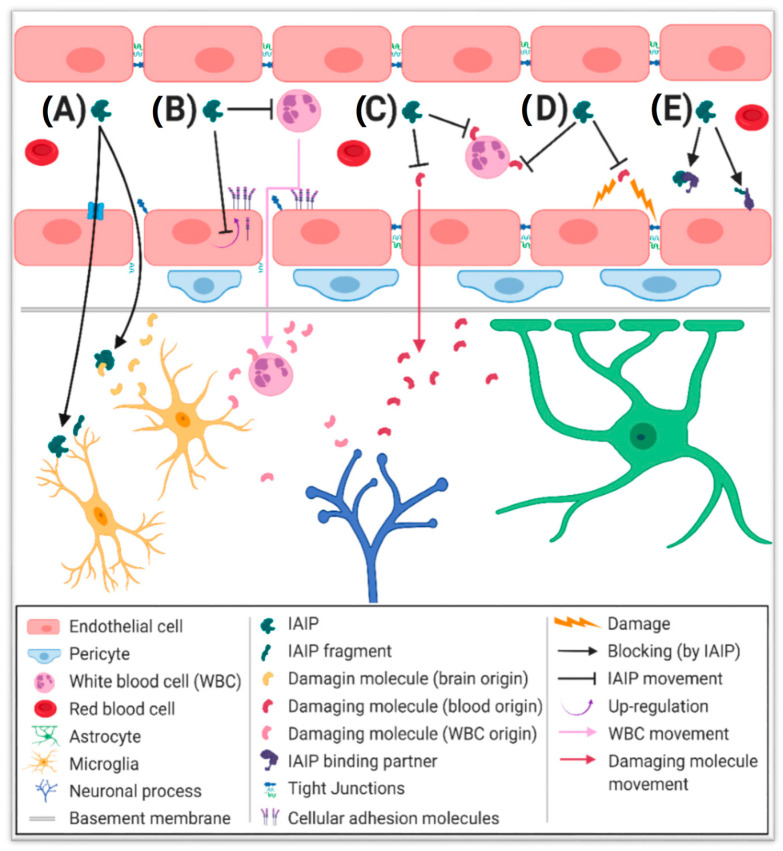
Stylized schematic of potential mechanisms underlying the neuroprotective properties of inter-alpha inhibitor protein (IAIP) treatment after exposure to hypoxia-ischemia (HI) in neonatal subjects. Legends for the elements in the Figure are illustrated below the Figure. The blood-brain barrier (BBB) is shown with endothelial cells joined by tight junction proteins. Red blood cells and white blood cells are depicted in the bloodstream. Pericytes line the vascular endothelium below the basement membrane. Astrocytes have endfeet that line the neurovascular unit, microglia, and neuronal processes are also in close proximity to the vasculature. The potential actions of IAIPs are also shown: (**A**) IAIPs or smaller subunit components of IAIPs could cross the BBB through paracellular transfer that is increased after HI damage or through yet to be identified transcellular pathways. Once within the central nervous system (CNS), IAIPs or IAIP components could interact with molecules or cells within the brain directly; (**B**) IAIPs might attenuate the transfer of activated immune cells into the brain by limiting the up-regulation of adhesion molecules on endothelial cells, altering the ability of immune cells to cross the vasculature or transforming immune cells to a non-damaging phenotype; (**C**) IAIPs may limit the transfer of harmful molecules into brain tissue by either directly inactivating the molecule or by preventing its production, release, or BBB passage; (**D**) Through similar mechanisms in (**C**), IAIPs might decrease the amount of harmful molecules in the systemic circulation, thereby preventing those molecules from damaging the integrity of the BBB; (**E**) IAIPs or components of IAIPs may form molecular complexes that either cross the BBB, line the BBB vasculature, or have downstream effects independent of the original IAIP molecule administered. Although the potential mechanism(s) of action of IAIPs depicted in (**A**–**E**) have some support, as summarized above, many of the specific molecular mechanism(s) at the BBB endothelium require future investigation. Cellular and molecular size, distribution and neurovascular coverage are not necessarily accurately depicted to allow for illustration of the potential mechanism(s) of action of IAIPs. Illustration was created with BioRender.com.

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
