# Peer review of "Novel Neuroprotective Agents to Treat Neonatal Hypoxic-Ischemic Encephalopathy: Inter-Alpha Inhibitor Proteins"

_ijms, 2020, doi:10.3390/ijms21239193_

Round 1
Reviewer 1 Report
The manuscript provided by Koehn et al. describes the neuroprotective effects for the hypoxic-ischemic encephalopathy of Inter-alpha inhibitors proteins. The manuscript is well written and summarizes relevant and novel aspects of the use of these proteins as therapy, The subject is very interesting and solid. Due that comments are minimal.
Minor Points,
page 2........is the only clinically approved neuroprotective treatment strategy for newborns exposed to HI.... (Please, provide a reference)
page 2......The amount of brain tissue preserved during and
directly after exposure to HI is a major factor in determining the extent of long-term cognitive deficits.
They could clarify or address how the amount of preserved brain tissue is evaluated, the phrase could be replaced by "severity of damage".
Author Response
Thank you very much for your kind comments. Both minor comments have been addressed in the revised manuscript:
Page 2: reference required
- Page2, paragraph 2, line 1: reference added. Note that all other reference numbers are now altered to accommodate the new citation.
Page 2: sentence clarification required
- Page 2, paragraph 2, line 4: sentence was removed to improve the clarity of the paragraph.
All changes to the original manuscript are highlighted by "track changes".
Reviewer 2 Report
In the manuscript entitled ‘Novel neuroprotective agents to treat neonatal hypoxic-ischemic encephalopathy: Inter-alpha inhibitor proteins’, the authors Koehn et al., have discussed the use of Inter-alpha inhibitor proteins (IAIPs) as potential therapeutic agents to treat neonatal hypoxic-ischemic encephalopathy. The review is well researched and detailed. The authors have very clearly summarized the literature including properties of IAIPs as neuroprotective agents in animal models and their mode of action. Further, they have discussed future directions that require substantial data on the efficacy and dosage of the IAIPs molecules. Overall this review has been thoroughly written and makes a strong case to consider IAIPs as promising new treatment options for neonatal hypoxic-ischemic encephalopathy.
Author Response
Thank you for you kind comments.